# Gender Differences in the Effects of BMI on School Bullying and Victimization in China: Comparing Normal Weight, Underweight and Overweight Secondary School Students

**DOI:** 10.3390/children9091388

**Published:** 2022-09-14

**Authors:** Yang Xie, Xiying Wang, Xiaotao Wang, Liu Liu

**Affiliations:** 1Institute for Education Theories, Faculty of Education, Beijing Normal University, Beijing 100875, China; 2School of Social Development, Nanjing Normal University, Nanjing 210023, China; 3School of Social and Behavioral Sciences, Nanjing University, Nanjing 210023, China

**Keywords:** Body Mass Index (BMI), school bullying, gender, sexuality education, China

## Abstract

Weight-related school bullying and victimization have become important public health issues among adolescents around the world. This study aims to examine gender differences in the effects of Body Mass Index (BMI) on school bullying and victimization among secondary school students. This study conducted a survey among 2849 adolescents—1393 girls (48.9%) and 1456 boys (51.1%). The students were between 12 and 18 years of age and were recruited from ten secondary schools in 2019 in Suqian City in China. The study showed that overweight boys were more likely to bully others and be bullied by peers compared to normal weight boys. In contrast, overweight girls reported less bullying than normal-weight girls. No significant relationship was found between overweight and victimization among female students. The implications for comprehensive sexuality education practices are also discussed.

## 1. Introduction

### 1.1. Sex Education in China: Situation and Challenges

This study aims to examine gender differences in the effects of Body Mass Index (BMI) on school bullying and victimization among secondary school students. In the study, BMI, a particular index of fast changing youth bodies, is selected and linked with diversified body images and potential risks of school bullying. The study aims to contribute to the field of sex education and school bullying.

In China, only 40% of young people have received some form of school sex education nationwide [1]. The term “some form of” refers to the fact that most students only attended fragmentary lectures on abstinence, puberty, hygiene and prevention of STD and HIV [2]. This type of sex education is limited and less effective. It emphasizes the knowledge of biological change, but fails to foster young students’ critical thinking in reflecting on the relations between changing body, gender-based body ideals and related consequences, such as exclusion and bullying. At the same time, it also fails to improve students’ skills in appreciating the beauty of various body shapes and establishing healthy relationships with peers. It often regards students as passive victims rather than active agents who can take control of their body and sexuality and make responsible choices. Therefore, it is crucial to provide sex education for adolescents to help them accept and appreciate their own bodies while respecting the uniqueness of others.

There is in urgent need to transform the current practice of school-based sex education into gender and rights-based comprehensive sexuality education (CSE), which has been approved as a much more effective model to equip young students with the right knowledge, skills and values regarding their sexuality, in order to help them appreciate their own uniqueness and build and maintain healthy relationships with others. In this light, this study is an endeavor to link bodily development, gender difference, and violence and bullying prevention in order to provide some implications for sex education practice in China and abroad. Moreover, this study is significant in contributing to the understanding of the three key concepts highlighted in the international technical guidance on sexuality education (revised version) including [3]: understanding gender (Key Concept 3), violence and staying safe (Key Concept 4), the human body and development (Key Concept 6).

### 1.2. Youth, Body Change and Gendered Body Ideals

Adolescence is a critical period for bodily development and physical changes. During this stage, adolescents incline to pay more attention to the changing bodies of their own and others. They are also more aware of how others evaluate their bodies and take a judgmental position of other people’s bodies. Therefore, their body anxiety, discomfort and dissatisfaction inevitably increase during this stage [3,4].

Body ideals are gendered and influenced by mass media and peers. Around the world, the most popular female body ideal is slim [5]. Mass media portrays women and girls as having svelte physiques with small waistlines and low body fat, i.e., the “thin ideal” [6]. Body perfectionism is not exclusive to girls. In fact, boys are also found to be under sociocultural pressure to achieve ideal body shapes [7]. The ideal male body is strong, with a v-shaped figure to show their masculinity [8]. Studies have pointed out that most secondary school students internalize the prevalent body ideals, with girls wanting to stay slim [9] and boys wishing to increase their muscularity [10].

All these internalized body ideals put certain groups of students, especially those who do not conform to ideal body shapes, at greater risk of appearance teasing and bullying [11]. A UNESCO report pointed out that physical appearance is one the most common reasons that students are bullied: 15.3–25% percent of bullied students reported that they were made fun of because of how their face or body looked [12]. In Asia, physical appearance is the leading cause of bullying among girls (19.2%), who are twice as likely to report it as boys (9.8%) [13]. Several longitudinal and cross-sectional studies have shown that female adolescents report higher levels of body dissatisfaction than male adolescents and overweight girls receive more negative feedback from friends [14]. Body dissatisfaction in adolescents can lead to many negative outcomes, such as substance use [15], low self-esteem [16], depression [17], body dysmorphic disorder [18] and eating disorders [19]. Body dissatisfaction can also induce school violence and bullying. Under the influence of body ideals, boys and girls with different body shape, size and image may have different experiences. This study uses students’ BMI as a classification criterion of their body shape in order to compare their experience of school bullying and capture gender differences.

### 1.3. BMI and School Bullying

School bullying is a worldwide social problem and China is no exception. It is a form of school violence characterized by repeated, intentional and aggressive behavior towards the victim and an imbalance of power between the bully and the victim [12]. In 2019, a report showed that 32% of students had been bullied by their peers at least once in the past month and 32.4% had experienced physical assault at least once in the past year [13]. School bullying is also considered a serious social problem in China [20]. Prior studies have found that the prevalence of victimization and perpetration of bullying in Chinese schools ranged from less than 5% to almost 70% [21].

School bullying is associated with numerous factors. In terms of individual factors, being overweight, lower socioeconomic backgrounds, having few friends and abusive experience are predictors of children and adolescents being involved in school bullying [22,23]. What is more, the prevalence of school bullying tends to decrease as adolescents age [24]. On the family level, good parental support and care may reduce the likelihood of bullying victimization [23]. Additionally, in China, the urban-rural distinction is a factor that cannot be ignored. Compared to urban adolescents, rural adolescents are more vulnerable to various types of school bullying [25].

Globally, weight-related school bullying and victimization have become important public health issues among adolescents [26]. Existing research has linked BMI and school bullying among adolescents. BMI, namely body mass index, is a criterion for weight classification, with three categories, underweight, normal weight and overweight. Previous studies found that overweight and underweight adolescents are more likely to be victims of bullying than their normal-weight peers [27,28]. They are also more likely to suffer from repeated bullying [29]. Adolescents who are dissatisfied with their bodies are more likely to become passive or reactive victims [30], especially if they are overweight. A longitudinal study showed that approximately a quarter of adolescents reported being teased about their weight during early and mid-adolescence and the rate of being teased was higher among overweight adolescents [31]. Even adolescents with a normal BMI but a false perception of their own weight are more likely to be teased and bullied [27]. Overweight students are more likely to be cyberbullied. Compared to other groups, overweight students are also more likely to bully others, both physically and relationally [32].

Existing studies have yielded inconsistent results on gender differences in the effects of BMI on school bullying. Some studies have shown that both being underweight and overweight are the predictors of victimization of bullying for both boys and girls [32,33]. However, other studies have found that BMI only has a significant effect on boys’ experience of bullying, such that overweight boys are more likely to be bullied and bully others [34] and underweight boys are more likely to be victims of bullying [35], but different BMI categories are not significantly correlated with girls’ experience of bullying. It has also been demonstrated that overweight boys are more likely to be perpetrators than normal weight peers [36,37] as they have a physical advantage over others [38].

The theory of masculinity and femininity is helpful for understanding weight-related school bullying. Masculinity and femininity are societal expectations that are used to distinguish men and women [39] and are the result of interaction and flexible constructs within the social environment [40]. As boys grow up, they are constantly expected to be strong and powerful, while avoiding feminine behaviors [41]. On the other hand, girls are constantly expected to be passive, tender and kind with thin and slim bodies. Collectively, youths grow up internalizing that only those who conform to the norms of masculinity and femininity are real men and women [42]. In contrast, violation of the norm sometimes leads to problems, including violence, aggression [43], bullying and victimization [44].

However, weight-related school bullying remains an understudied phenomenon in China. Therefore, it is meaningful to investigate the relationship between BMI and school bullying and victimization in China among both male and female students. Based on the above literature review, this study proposes the following research hypotheses: (1) Overweight boys are more likely to bully others and overweight boys are more likely to be bullied by others. (2) Both overweight girls and underweight girls are more likely to be bullied and less likely to bully others.

## 2. Materials and Methods

### 2.1. Sampling and Procedure

In this study, a multi-stage whole-group sampling method was used to choose 10 randomly-selected schools in Suqian City, Jiangsu Province, China—five middle schools, four high schools and one vocational school. In each school, two classes were randomly selected from classes (graduating classes were excluded) and all students in the selected class formed the sample. The sample includes 2849 adolescents: 1393 girls (48.9%) and 1456 boys (51.1%). The Faculty of Education, Beijing Normal University, reviewed and approved the research protocol of the survey before data collection took place in early 2019. Administrators of all secondary schools in which the study’s data was collected also acknowledged the participants’ protection under the Human Subjects Protocol.

All participants were informed of the purpose and content of this study. All the participants were surveyed anonymously in their own classrooms through the paper version of the questionnaires. Only the researchers were present in case the participants needed clarification. Teachers were excused to ensure the participants’ privacy and anonymity.

### 2.2. Measures

Considering the methods used in previous studies to measure participants’ BMI, such as self-administered questionnaires, direct weight measurement, etc., in this study, respondents’ height and weight were self-reported to the exact size in cm and weight in 0.1 kg. BMI was then calculated using the formula weight (kg)/height (m)^2^. All adolescents were classified into three groups based on their height and weight: underweight, normal weight and overweight. The classification criteria of BMI in this study were based on the screening for overweight and obesity among school-age children and adolescents and the screening standard for malnutrition of school-age children and adolescents published by the National Health Commission of the People’s Republic of China [45,46], in which adolescents aged 7 to 18 years are classified according to different distinct criteria and there are differences in the metrics for boys and girls. For example, for a 12-year-old boy, BMI over 20.7 is considered overweight and BMI less than 15.4 is considered underweight; for a 12-year-old girl, BMI over 21.5 is considered overweight and BMI less than 14.7 is considered underweight; while for an 18-year-old boy, BMI over 24.0 is considered overweight and BMI less than 17.9 is considered underweight; for an 18-year-old girl, BMI over 24.0 is considered overweight and BMI less than 17.3 is considered underweight.

The Chinese version of the Illinois Bully Scale (IBS) was employed to assess the participants’ experiences of bullying and victimization within the past 30 days [47]. Particularly, data from the bullying subscale and the victimization subscale were used. The bullying subscale contains nine items, such as “I have hit students who are easier to bully”. The victimization subscale contains four items, such as “Other students have hit me or pushed me”. They were both Likert-type scales with responses ranging from 0 to 4 (“never”, “one to two times”, “three to four times”, “five to six times” and “seven or more times”). Therefore, scores for the bullying subscale range from 0 to 36 and the range of scores for the victimization subscale is 0 to 16. The higher the score for the respondents on the IBS, the higher the tendency towards involvement in school bullying behaviors, either as perpetrators or victims. In this study, the IBS had high internal consistency (bullying: Cronbach’s α = 0.80; victimization: Cronbach’s α = 0.77).

Several sociodemographic and other variables were included in this study to form control variables. Considering the predictors of school bullying that have been verified by previous studies, this study included age, number of friends, whether or not they had experienced abuse (both physical and emotional) and parental care as control variables. We used a single question, such as “Have you ever been abused?” to measure abuse history. Response categories ranged from “never been abused”, “physically abused”, to “mentally abused”. The item was coded as 0 = no and 1 = yes. The paternal care subscale and maternal care subscale of the Chinese version of the Parental Bonding Instrument (PBI) were used to measure participants’ parental care. The questions in the scale, such as “treats me with affection”, contain four options, 0 = ”very much not true”, 1 = ”rather not true”, 2 = “more consistent” and 3 = “very consistent”. The higher the participant’s score the more parental love they feel. Since mass media is the main source of information influencing the body ideals of adolescents, we also included weekly Internet time as a control variable. In addition, considering the special circumstances of China’s test-oriented education system and urban–rural differences, we also included the place of household registration and passing/failing exams as control variables in the analysis.

### 2.3. Statistical Analysis

STATA software 14.0 (StataCorp, College Station, TX, USA, 2015) was used for statistical analysis in this study. First, descriptive statistics were performed on the sample data to examine the BMI and bullying of the sample. Secondly, the model was divided into two parts, the male and the female sample, and robust regression analysis examined the relationship between BMI and school bullying separately.

## 3. Results

### 3.1. Descriptive Statistics

Table 1 presented the statistical analyses of the demographic information of 2849 participants who were either underweight (9.23%), normal weight (75.29%), or overweight (15.48%). The age of the participants ranged from 12 to 18 years old, with an average age of 14.9 years and the majority of the participants were from urban areas (60.58%). 30.99% of participants had failed at least one exam, with a higher percentage of boys than girls. Participants had an average of seven friends and their average weekly Internet time was 11.3 h. Some participants reported experiences of physical (9.09%) and mental (14.39%) abuse, with boys self-reporting more physical and emotional abuse. What is more, 52.4% of participants had experienced bullying or victimization three to four times in the past 30 days.

In terms of BMI, the proportion of underweight, normal weight and overweight male students were 13.19%, 66.14% and 20.67%, respectively; the corresponding values for female students were 5.1%, 84.85% and 10.05%. In terms of bullying, male students’ average scores on bullying and victimization scales were 3.57 and 3.40, while female students’ average scores were 2.65 and 2.49. The results of the independent samples t-test showed that the difference between boys’ and girls’ scores on the bullying subscale and the victimization subscale was significant, implying that male students are more likely than female students to be involved in school bullying.

### 3.2. Weight and Bullying: Gender Differences

The sample was divided based on gender and analyzed separately. The scores on the bullying scale (bullying and victimization) were entered as the dependent variable and the BMI was entered as the independent variable and variables like age and weekly internet time were included as a set of control variables. A robust regression analysis was then conducted (Table 2).

Table 2 showed that BMI significantly predicted students’ involvement in school bullying and showed significant gender differences. For male students, overweight boys reported more bullying (Co = 0.37, *p* < 0.05) and were also more likely to be victims (Co = 0.38, *p* < 0.05) compared to normal-weight boys. A significant effect of weight group on bullying was revealed for female students but not for victimization. Overweight girls were less likely to be bullies than normal-weight girls (Co = −0.47, *p* < 0.05). In addition, passing/failing exams and weekly Internet time had a significant effect on bullying and victimization for girls, while experiences of abuse and parental care had a significant effect on bullying and victimization for both boys and girls.

## 4. Discussion

### 4.1. Findings

Weight-related school bullying and victimization remain a severe concern in China. The different body expectations of boys and girls lead to gender differences in weight-related school bullying. The current study shows the significant gender differences in the effects of BMI on school bullying and victimization among Chinese secondary school students and seeks to provide an empirical basis for the future development of comprehensive education (CSE) in China. The present study found that overweight boys were more likely to bully others and be bullied by their peers than normal weight boys, partially validating hypothesis (1) and consistent with previous research [30]. In contrast, overweight girls were less likely to bully others than normal weight girls and no significant relationship was found between being overweight and bullying of girls, which partially verified hypothesis (2).

Overweight boys were more likely to be bullies, while overweight girls instead reported less bullying than normal weight girls. The difference between overweight boys and overweight girls may be explained by the theory of masculinity and femininity. Boys are expected to behave in accordance with social standards that restrict them from exhibiting behaviors associated with femininity [48]. Physically strong boys tend to have a high status, while thin boys are considered weak and lacking in masculinity [49]. Particularly for boys who have higher BMI scores due to muscle mass and athletic physique, physical dominance gives them greater power over their peers, making them more likely to bully others. In contrast, girls aspire to have slim figures [50]. Therefore, overweight girls will be more interested in becoming thin rather than exploiting their physical superiority to bully others.

Notably, this study found no significant relationship between overweight and victimization of female students, which is inconsistent with the results of most existing studies [49]. There is no doubt that Chinese female adolescents also identify a slim body as the ideal body, but why this body tendency does not raise the issue of bullying overweight girls is worth exploring. In the Chinese context, grades constitute a predominant source of stress for secondary school students. Some studies have shown that overweight adolescents do not differ significantly from normal-weight adolescents in terms of academic achievement [51]. Moreover, the control variable of passing exams or not had a significant effect on girls’ victimization in this study, with girls who received a passing grade and above on all tests being less likely to be victimized. Therefore, a strong focus on grades may diminish the impact of weight on school bullying. These findings diverge from prior studies in the literature in English and highlight the importance of considering the impact of cultural and historical factors when attempting to understand the linkage between bullying and body weight among adolescents.

### 4.2. Limitations and Future Studies

The potential limitations of this study need to be noted. First, the BMI data were calculated using participants’ self-reported height and weight. In this case, the participants’ self-reports may differ from the actual situation, which in turn may affect the study results. At the same time, the use of BMI alone as a basis for classification may be inadequate. Future studies could use more accurate instruments to measure participants’ height and weight directly in order to obtain more accurate BMI data. In addition, waist-hip ratio, bioimpedance measures or skinfolds should be considered when defining underweight or overweight, rather than BMI alone.

Second, this study used cross-sectional data, which can only test the correlation between variables and cannot further verify the causal relationship. Future research could adopt time series data to further validate the mechanism of occurrence of weight-related school bullying.

Third, the statistical data in this study were collected in one city in China, which may be subject to selectivity bias, and the sample may be underrepresented. In particular, no correlation between overweight and female student victimization was found in this study, which may be related to both the test-based education system in China and the underrepresentation of the sample. Studies in multi-city, cross-regional and multicultural contexts should be conducted in the future.

Fourth, this study did not explore the intervention effects of sex education on weight-related school bullying. Future studies could use a randomized controlled trial (RCT) to validate the effect of sex education on the relationship between BMI and school bullying.

### 4.3. Implications

Based on the results of this study, the implications for future practices are as follows. Because weight-related bullying in school is closely related to body ideals based on traditional gender norms, comprehensive sexuality education is encouraged to help adolescents develop critical thinking about socially constructed gender norms.

Body image is an effective way to prevent and reduce weight-related bullying in school. Body image is a core concept in sex education programs for adolescents and is an important topic in the international technical guidance on sexuality education. For adolescents aged 12 to 18, sex education focuses on making them aware that how people feel about their bodies affects their health, self-image, behavior and unrealistic standards of physical appearance should be challenged [12].

In developing sex education practices for adolescents, particular attention should also be paid to gender differences in body images and traditional gender norms among adolescents and differential sexuality education strategies should be implemented for boys and girls. For boys, CSE programs that involve discussions of masculinity should gain the attention of practitioners and policymakers. In fact, many experiences of bullying are rooted in or explained through masculinity [52] and body image is one of the manifestations of masculinity. However, many sex education programs lack a discussion of masculinity because masculinity is often considered customary and reasonable, yet boys do not feel that these sex education programs address their needs and questions regarding sexuality [12]. Implementing a CSE program that includes a discussion of masculinity, not only to make boys aware that a person’s physical appearance affects how others feel and act toward them but also to help boys analyze specific cultural and gender stereotypes [12] and thereby break down the obsession with masculinity, can have a positive impact on preventing bullying in schools and promoting gender equality.

For girls, the focus of sex education programs should be on CSE curricula that focus on gender, power and critical thinking to help them critically assess gender standards of beauty. Although no significant relationship was found between being overweight and the victimization of girls in this study, globally, girls are more likely than boys to experience appearance-based bullying and are more often made fun of for their appearance [12]. Girls’ perceived pressure on appearance is closely related to traditional gender norms and an empowerment or rights-based approach to CSE interventions that include elements of gender norms, power relations and rights can have a positive impact on girls [3]. By analyzing common practices that people use to try to change their appearance and assessing the dangers of these practices, girls are made aware that using drugs to change body image can be harmful. In addition, by analyzing specific cultural and gender stereotypes and how they affect people’s body image and their relationships, girls can be made aware that unrealistic standards for physical appearance can be harmful, which in turn can help them reflect on their body image and challenge traditional gender norms [3]. Therefore, conducting empowering CSE programs focused on gender, power and critical thinking can be an effective approach to changing girls’ gender perceptions and reducing weight-related school bullying.

## 5. Conclusions

This study contributes to the understanding of the gender difference of weight-related bullying and victimization in China. The implementation of differentiated sex education strategies for boys and girls is also discussed. This study points out that the paradigm of current fragmental sex education need to be upgraded into a Comprehensive Sexuality Education model that focuses on raising teenagers’ gender awareness, encouraging critical thinking and empowering them to be active agents in making responsible decisions.

## Figures and Tables

**Table 1 children-09-01388-t001:** Descriptive Statistics for Male (*n* = 1456) and Female (*n* = 1393).

	All	Male	Female	*p*
Bully	3.12	3.57	2.65	***
Victim	2.96	3.40	2.49	***
BMI				***
Underweight (%)	9.23	13.19	5.10	
Normal weight (%)	75.29	66.14	84.85	
Overweight (%)	15.48	20.67	10.05	
Age	14.91	15.01	14.81	***
Household registration				n.s.
Rural (%)	39.42	39.84	38.98	
Urban (%)	60.58	60.16	61.02	
Pass exams or not				**
Yes (%)	69.01	66.69	71.43	
No (%)	30.99	33.31	28.57	
Number of friends	7.28	8.49	6.02	***
Weekly Internet time	11.39	12.29	10.45	***
Abuse				
Physical abuse				***
Yes (%)	9.09	12.02	6.03	
No (%)	90.91	87.98	93.97	
Mental abuse				**
Yes (%)	14.39	16.55	12.13	
No (%)	85.61	83.45	87.87	
Mother care	22.78	22.71	22.86	n.s.
Father care	21.48	21.41	21.55	n.s.

Note: **, *p* < 0.01; ***, *p* < 0.001. n.s., *p* > 0.05 and *p* refers to the tests for gender difference.

**Table 2 children-09-01388-t002:** Robust Regression Predicting Bully and Victim for Male (*n* = 1456) and Female (*n* = 1393).

Independent Variables	Model 1-1		Model 1-2		Model 2-1		Model 2-2	
	Bullying (Male)		Bullying (Female)		Victimization (Male)		Victimization (Female)	
	Coefficient	s.e.	Coefficient	s.e.	Coefficient	s.e.	Coefficient	s.e.
Cons.	6.81 ***	1.50	1.63	1.41	5.70 ***	1.56	3.77 **	1.21
BMI (Normal weight = 0)								
Underweight	0.31	0.21	−0.39	0.27	0.23	0.22	−0.27	0.24
Overweight	0.37 *	0.18	−0.47 *	0.20	0.38 *	0.19	0.15	0.17
Age	−0.16	0.10	0.187	0.10	−0.12	0.10	0.05	0.08
Household registration (Rural = 0)	−0.09	0.17	−0.02	0.15	−0.02	0.18	−0.12	0.13
Pass exams or not (No = 0)	−0.26	0.16	−0.27	0.15	−0.20	0.17	−0.56 ***	0.13
Number of friends	−0.01	0.00	−0.01	0.01	−0.00	0.00	−0.00	0.00
Weekly Internet time	0.01 *	0.00	0.02 **	0.00	0.01	0.00	0.01	0.00
Abuse								
Physical abuse (No = 0)	0.20	0.25	0.25	0.29	1.68 ***	0.26	−0.18	0.24
Mental abuse (No = 0)	1.30 ***	0.22	0.56 **	0.21	0.83 ***	0.23	1.24 ***	0.18
Mother care	−0.05 ***	0.01	−0.02 *	0.01	−0.04 *	0.01	−0.04 ***	0.01
Father care	0.00	0.01	−0.02	0.01	−0.018	0.01	−0.02 *	0.01
R-Squared (%)	13.70		12.21		14.95		15.80	

Note: *, *p* < 0.05, **, *p* < 0.01; ***, *p* < 0.001. Dummy variables representing the schools are also included in the models.

## Data Availability

Ethics approvals do not permit these potentially re-identifiable data to be made publicly available.

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
