# Peer review of "Gender Differences in the Effects of BMI on School Bullying and Victimization in China: Comparing Normal Weight, Underweight and Overweight Secondary School Students"

_children, 2022, doi:10.3390/children9091388_

Round 1

Reviewer 1 Report

The manuscript addresses an interesting and relevant topic which links weight related bullying among adolescents using a sample from China.

In general, several issues need to be adequately addressed before the manuscript can be considered for publication:

Introduction: Although the study uses data from China, the issue is relevant for an international audience. The authors should consider their results implication for sex education internationally, not only in China, as stated in lines 61-63. My concern is, if the focus is on sex education in China, then we need a background/context of sex education in China, as how it is organised, curriculum, implementation and so on. Meanwhile, in lines 49-51 mentions ‘boyish girls, feminine boys’ which brings up another difficult issue when it comes to sex education in China, as the Chinese government dose not officially recognise ‘LGBT’, we wonder if and how these issues are addressed in sex education in China. As the manuscript title suggested, this study focuses on BMI and school bullying and their relations to genders, which may be more relevant to school education in general than to sex education in the case of China. However, later on in Literature Review, lines 125-144 provide some background/context needed which should be expanded and moved to Introduction before the texts of lines 61-62 mentioning CSE in China. Some critiques or rooms for improvement of CSE in China based on an international standard here would support more the discussion later on.

Literature review: lines 82-83 mentioning ‘Northwestern Countries’ is confusing as which parts of the world are meant as the texts followed with reference nr.21 is a study from Malaysia.

3.2 Measures: Lines 200-202 mention ‘questions on whether they had enough care from the parents’ which needs more description in order to understand the numbers reported as mother care and father care in Table 1 descriptive statistics.

4.2 Weight and bullying: Gender differences: Table 2 appears non-complete as we only see victim (male) model,  not victim (female) model, and we wonder why the other groups are not included in the analysis? As mentioned in section 3.2 lines 196-197 that “the sample can be classified into four types: bullies, bullied, bully-bullied, and neither bullying – nor bullied. Inconsistency needs to be correct between Table 1 and Table 1 as ‘Household registration’ in Table 1 becomes ‘Hukou’ in Table 2. Lines 227-230 mentioning ‘numbers of friends’ as control variable, which measures is not described in 3.2 nor in Table 1 while Table 2 presents “number of same-sex friends’. We wonder why using “same-sex friends” and not all friends? Moreover, why Table 2 presents “Age and number of friends” the same way as other variables since they are ‘control variables’ as mentioned in lines 227-230? By the way, why should these two be control variables?

Discussion and 6. Conclusion: Table 2 is not complete so it is difficult to follow the discussion. Moreover, it is difficult to relate the results and discussion to CSE as a solution to this weight-bullying problem. Even though it was good result from a previous experiment (as mentioned in the texts), more evidence from elsewhere in the world would strengthen the authors' argument and suggestions of an improved CSE in China with international standard.

The manuscript needs a round of language editing before re-submission.

Reviewer 2 Report

The manuscript entitled "Gender Differences in The Effects 0f BMI on School Bullying and Victimization in China: Comparing Normal Weight, Underweight, and Overweight Secondary School Students" with ID " children-1850756", overall it is an interesting study conducted with a large sample of Chinese participants on one of the most relevant public health problems worldwide: bullying. I think that the authors will be able to make some changes. My main concern lies in the introductory section and the method used to explore the relationship between BMI and bullying.

Therefore, below I develop comments in relation to the present manuscript.

1.       Throughout the manuscript, the authors talk about "gender differences", however, after reading it I think they are referring to “sex differences”. This fact should be clarified. Have the authors used information about the biological sex of the participants (girls vs boys) or on the contrary have they collected information about their gender (male, female, non-binary...).

2.       ABSTRACT: Around the globe? I feel it is better to said around the world or worldwide.

3.       Regarding the INTRODUCTION: Firstly, I believe that the Introduction and literature review sections should be merged into a single Introduction section. In it, the authors could highlight the importance of their research as well as the background found in previous evidence. The most important concern is that the introduction should be improved. Although it gives importance and background to the problems of overweight during adolescence, there is a lack of information about the problem of bullying (e.g. prevalence of bullying, definition, roles they can take). Furthermore, the authors should also provide more information on sex differences associated with this behavior. It would also be useful to add information on other studies that may have used other measures to assess overweight (folds, bioimpedance studies, etc.).

4.       The association between BMI and bullying focuses only on victimization. The authors explore the association that BMI presents with the role of bully. No previous studies explored the association between BMI and other bullying roles (bully, bully/victim)? As bullying is a type of aggression, studies that explore the relationship between aggression and bullying could be mentioned.

5.       In the introduction between lines 39 and 42, the authors write two sentences, which are repetitive.

6.       The last paragraph of the introduction gives information about implications not about the background of the study.

7.       The information reflected between lines 128 and 144 on sex education should be omitted from the Introduction section. If the authors are not going to explore the role of sexuality education in their work, I think this would be better reflected in the implications section.

8.       OBJETIVE: The sentence “Considering the differences in traditional gender norms of masculinity and femininity required for boys and girls, it is worth investigating the relationship between BMI and school bullying and victimization in China among both male and female secondary school students. Moreover, the results of this study can help provide an empirical basis for conducting CSE in China” is confusing regarding the objective of the study.

9.       MATERIAL AND METHODS: Which is the city in which the study was carried out?

10.   MATERIAL AND METHODS: The authors claim to have based themselves on the criteria of the National Health Commission of the People's Republic of China, but what are the cut-off points for this classification? I think that for readers who do not check this classification, this information should be expanded. What BMI values were considered normal weight? In this line, it is surprising that there are only three categories (underweight, normal weight and overweight) as the WOS classification differentiates very well in the overweight group to be overweight, obese or morbidly obese.

11.   MATERIAL AND METHODS: Other of the principal concerns is the method used to assess bullying. First, information about example items for each of the subscales would help to better understand the questionnaire. Furthermore, the authors show that thanks to this scale four types of roles can be identified: bullies, bullied, bully-bullied, and neither bully-nor bullied. I would rewrite them because of how they have been mentioned in previous works: not involved in bullying, victims, bullies, or bullies/victims. It is surprising that they may not have considered all three roles in the study. What reason is there to explore only the role of victim and bully?

12.   MATERIAL AND METHODS: The last paragraph in which information on covariates is included is confusing. First, it is the first time they are mentioned and they are not correctly justified in the introduction. To continue, only two are cited (household and internet usage). Nevertheless, the sentence starts by saying "several covariates were used..." were only these two used? If there are more, they should be identified here and the questionnaires/materials used to assess them and how they are scored should be explained. Furthermore, this should be justified in the introduction beforehand.

13.   RESULTS: Again, giving information about the descriptives of many measured variables is surprising as it is the first time they are mentioned and it is not clear why they have been measured. In case it is only to have the characteristics of the sample, this should be justified. Otherwise, it seems that the authors put this information to show everything they have collected.

14.   RESULTS: In the method section, it indicates that due to the cut-off point, the students can be identified and classified into one of the 3 roles (victim, bully and bully-victim), however, in this section the continuous variable is presented. I consider that the continuous variable does not provide sufficient information because if the participant scores 0 or 1, he/she is not considered to be involved and could be raising the mean. I consider that for this type of work it is better to use the categorical variable and explore how BMI is related to bullying involvement on a role-by-role basis. I would not ignore the readers of the categorical variable, because of the information it provides.

15.   RESULTS: I miss data on prevalence of bullying. Yet again, this could have been calculated in a simple way using percentages and bullying frequencies (with the categorical variable).

16.   RESULTS: The descriptive table showing differences by sex, information about the test performed (is it a t-test or non-parametric test) should also be provided. In addition, did the authors explore the normality of the variables? Information about the tests used.

17.   Because of the aforementioned, it would be very interesting to explore the role of stalker/victim and to run models using the variables in their categorical form. For this purpose, it would be advisable for the authors to use binary logistic or multinomial logistic regression models using continuous variables. Does the coefficient refer to the beta? In addition, I assume it will be standard error, but this should be indicated in the note to the table.

18.   RESULTS: Why is no information provided on the pattern of victimization of women? It should state that no direct effect of the variables explored was found and present the results.

19.   I think it is useful to unify the discussion points, limitations and implications.

20.   Limitations: Overall, this point is well drafted but perhaps the authors should highlight one more limitation. In assessing the effect of weight on bullying they focused on BMI data, however, waist-hip ratio, bioimpedance measures or even on skinfolds could be of interest.

21.   REFERENCES: Correct the reference 43.

Round 2

Reviewer 1 Report

I want to thank the authors for a much revised manuscript. However, several issues remain to be addressed before it can be considered of publishing.

I am not expert of this subject, however, as an ordinary reader, I think the authors should work a little more on: The relevance of sex education in bullying contexts needs more clarification, especially since there is no data supporting this relationship. I have suggested to the editor that the manuscript needs one more reviewer.

In Table 1 and Table 2 include variables 'age', 'household registration urban/rural', 'weekly internet use', but there is no discussion on the relevance and reason of including these variables in this study and there should be (e.g. Introduction or previous research, section 4. Materials and Methods.) Moreover, there should be some lines mentioning the absence or existence of effects from these variables and meaning in section 5 Results and 6 Discussion.

A minor point: the concluding sentence of Section 5 Results does not make any sense, i.e. lines 368-369: There was no significant difference between underweight students and normal-weight students.
